# Intracerebral Hemorrhage-Associated Iron Release Leads to Pericyte-Dependent Cerebral Capillary Function Disruption

**DOI:** 10.3390/biom15020164

**Published:** 2025-01-22

**Authors:** Stefanie Balk, Franziska Panier, Sebastian Brandner, Roland Coras, Ingmar Blümcke, Arif B. Ekici, Jochen A. Sembill, Stefan Schwab, Hagen B. Huttner, Maximilian I. Sprügel

**Affiliations:** 1Department of Neurology, Friedrich-Alexander-Universität Erlangen-Nürnberg, Schwabachanlage 6, 91054 Erlangen, Germany; 2Department of Neurosurgery, Friedrich-Alexander-Universität Erlangen-Nürnberg, Schwabachanlage 6, 91054 Erlangen, Germany; 3Department of Neurosurgery, Fürth Hospital, Jakob-Henle-Straße 1, 90766 Fürth, Germany; 4Department of Neuropathology, Friedrich-Alexander-Universität Erlangen-Nürnberg, Schwabachanlage 6, 91054 Erlangen, Germany; 5Institute of Human Genetics, Friedrich-Alexander-Universität Erlangen-Nürnberg, Kussmaulallee 4, 91054 Erlangen, Germany; 6Department of Neurology, Justus-Liebig-University Giessen, Klinikstraße 33, 35392 Gießen, Germany

**Keywords:** intracerebral hemorrhage, stroke, pericyte, iron, human brain, ferroptosis

## Abstract

Intracerebral hemorrhage leads to immediate brain injury due to local mechanical damage, on which current treatment approaches are focused, but it also induces secondary brain injury. The purpose of this study is to characterize blood components, degradation products and their effects in secondary brain injury. Immunocyto- and immunohistochemistry, Fluorescence-Activated Cell Sorting, WST-1 assays and RNA sequencing were applied using human cell cultures and human ex vivo brain tissue slices. Brain tissue was immediately collected, cooled and sliced during neurosurgical operations to perform experiments on living tissue slices of the human brain. Among the blood degradation products, free iron (Fe^2+^ and Fe^3+^), but not hemoglobin, induced detrimental effects on pericyte function and survival (78.5% vs. 94.3%; *p*-value < 0.001). RNA sequencing revealed ferroptosis as the underlining cellular mechanism, mediated via GPX-4 (log2 fold change > 1.0, *p*-value < 1.08 × 10^−30^) in pathway analysis and eventually resulting in oxidative cell death. Pericytes located at cerebral capillary branching sites were specifically affected by ferroptosis, leading to capillary disruption and vasoconstriction, which were partially prevented by ferrostatin-1. Free iron induces the pericyte-dependent disruption of cerebral capillary function and represents a therapeutic target to attenuate secondary injury after intracerebral hemorrhage.

## 1. Introduction

Intracerebral hemorrhage represents a devastating disease and largely contributes to the global burden of disease [1,2]. Treatment approaches in clinical practice currently aim to attenuate immediate brain injury caused by local mechanical damage [3]. However, intracerebral hemorrhage induces secondary brain injury, leading to further damage of brain tissue, substantially exceeding the immediate area of hemorrhage and resulting in worse clinical outcomes [4].

Clinical studies characterized perihemorrhagic edema formation by imaging scans and identified that, while the intracerebral hemorrhage gradually degraded, the surrounding edema correspondingly increased within 2–3 weeks after the initial bleeding event [5,6]. The extent of edema formation was associated with worse clinical outcomes [5,6]. Therefore, degradation products gradually released from the hematoma must be suspected to drive secondary injury in intracerebral hemorrhage.

Experimental studies focused on specific signaling pathways, molecules or proteins to identify potential therapeutic targets, but these highly specified research approaches have not resulted in successful treatment options for intracerebral hemorrhage [7,8,9]. While disruption of the blood–brain barrier has been determined as major factor of perihemorrhagic edema formation, and the crucial role of iron in the pathogenesis and progression of various diseases has been unraveled in recent years, the effect of immediate blood degradation products, such as iron, on blood–brain barrier function and its secondary injury in intracerebral hemorrhage has not been thoroughly investigated [10,11,12].

In this study, we assessed the effect of iron and hemoglobin on the survival and function of human capillary pericytes as essential cells for maintenance of the blood–brain barrier [13,14]. Next, transcriptomics analysis was used to ascertain which expression clusters were altered, thereby identifying ferroptosis as a crucial pathway that pericytes undergo when exposed to iron. We used human ex vivo brain tissue to demonstrate the occurrence of ferroptosis in pericytes at the capillary branching sites. Finally, we investigated the impact of iron exposure on vascular capillaries in ex vivo human brain tissue slices in terms of vascular continuity and vasoconstriction.

## 2. Materials and Methods

### 2.1. Drugs

All substances were diluted in water, as suggested by the manufacturer. Ammonium iron (II) sulfate hexahydrate (FAS, Ref No. 9719, Sigma-Aldrich, St. Louis, MO, USA), which we termed divalent iron, was applied at a concentration of 300 µM, if not explicitly stated otherwise. TDA Reagent for microbiology, which we termed trivalent iron (FeCl_3_, Ref No. 80353, Sigma-Aldrich), was applied at a concentration of 200 µM, if not explicitly stated otherwise. The rationale behind the selection of a slightly reduced dose of FeCl_3_ compared to FAS is due to the complexity of the integration of different experimental designs, with dose titration required to achieve a comparable level of common cell/tissue toxicity effects to FAS, as there has been no comparison of these two substances in this form to date. Deferoxamine mesylate salt (DFX, Ref No. D9533, Sigma-Aldrich) was applied at a concentration of 250 µM, ferrostatin-1 was applied at a concentration of 1 µM (Fer-1, Ref No. SML0583, Sigma-Aldrich) and both Hemoglobin A0 Ferrous stabilized human (Hb, Ref No. H0267, Sigma-Aldrich) and Hemoglobin human (Met-Hb, Ref No. H7379, Sigma Aldrich) were applied at a concentration of 9.3 µM. The substances and concentrations used were adapted from previous work [15,16,17,18,19,20].

### 2.2. Pericyte Cell Culture

Human brain vascular pericytes (InnoProt, Derio, Spain) were cultured in pericyte medium (InnoProt, Derio, Spain) containing 2% fetal bovine serum, 1% pericyte growth supplement and 1% penicillin/streptomycin solution. Cells were cultured on Poly-L-Lysine (2 µg/cm^2^, ScienCell) and incubated at 37 °C and 95% O_2_/5% CO_2_ until they reached confluence. Medium was changed every second day.

### 2.3. Cell Viability Assay

Human brain vascular pericytes were seeded into a 96-well plate (seeding density 5 × 10^5^/well) coated with Poly-L-Lysine. After 24 h, cells were incubated with FeCl_3_ (2, 20, or 200 µM), FAS (3, 30, or 300 µM), and/or DFX (250 µM) in pericyte medium for either 1, 2, 4 or 24 h. After incubation, cells were washed 3 times with PBS. WST-1 reagent (Roche) was applied to the cells at a ratio of 1:10 and cells were incubated for 4 h at 37 °C and 95% O_2_/5% CO_2_. Absorption was measured at 405 nm with Tecan Infinite M Plex Multiwell-plate-Readers (ELISA-Reader), and the wave length reference was 750 nm.

### 2.4. Flow Cytometry 

Live/dead staining was performed using VivaFix 583/603 Cell Viability Assay (Bio-Rad Laboratories, South Granville, NSW, Australia) according to the manufacturer’s instructions. Human brain vascular pericytes were seeded as described and afterwards incubated with FeCl_3_ (200 µM), FAS (300 µM), Hb (9.3 µM), Met-Hb (9.3 µM), and/or DFX (250 µM) in pericyte medium for 4 h at 37 °C and 95% O_2_/5% CO_2_. For flow cytometry staining, cells were detached, washed with PBS, and centrifuged at 300× *g* for 10 min at room temperature (RT). Cells were stained with VivaFix 1:500 diluted in PBS in the dark for 30 min at RT. After washing 2 times with PBS, cells were fixed with 4% PFA for 15 min. After 2 more washing steps with PBS, cells were eluted in FACS Flow Analysis Puffer (PBS containing 2% Fetal calve serum 0.2% EDTA 500 mM) for acquisition. Flow cytometry was performed with LSR Fortessa X-20 (BD FACSDivaTM, Software V.8.0.2). The software used for analysis was FlowJoTM V.10.8.1 (BD Biosciences, Franklin Lakes, NJ, USA).

### 2.5. RNA Sequencing

The “RNeasy Kit” from Qiagen was used to obtain total RNA from pericytes. Pericytes were treated with FeCl_3_ 200 µM in pericyte medium for 24 h. The following day, pericytes were collected, washed 3 times with PBS, and centrifuged at 300× *g* for 10 min at room temperature. Total RNA was extracted according to the manufacturer’s instructions. RNA sequencing, including RNA QC, library preparation, sequencing and raw data evaluation, was carried out in collaboration with the Next-Generation-Sequencing Core Unit in the Institute of Human Genetics at the University of Erlangen. Using the TruSeq Stranded mRNA Kit (Illumina, San Diego, CA, USA), a cDNA library was constructed and sequenced (101 bp), paired-end on a HiSeq2500 Platform (Illumina, San Diego, CA, USA). Raw data were converted into reads and saved with a quality score (bcl2fastq v2.17). The reads (averaging 37.7 million) were of very high quality with minimal quality trim (averaging 99.9% of raw data remaining). The STAR aligner (version 2.7.8.a) was used to map the data against the GRCh38 reference genome. Principal component analysis (PCA) of the top 500 genes showed good clustering. The unique mappings were then counted using featureCounts (version 2.0.1) according to the Ensembl 103 genes. Counting takes into account the strand specificity of the library preparation, allowing genes with different orientations to be counted, even if they overlap. Statistical analysis of the RNA sequencing data was performed using the DESeq2 R package version 1.30 in the R statistical environment version 4.0.3.

### 2.6. Cluster Analysis

A log2 fold change (log2FC) cut-off was set at −0.5/+0.5, with a cut-off for corrected *p* value ≤ 0.01. The Ingenuity Pathway Analysis (IPA) database from Qiagen was used to assign significant genes to anatomical–functional clusters. For selected clusters, a list of genes contained in the cluster was reproduced from the Ingenuity database. For graphical representation, heat maps were generated using the R program (ComplexHeatmap package). The heat maps show the top 70 genes of each cluster according to their absolute log2FC and marked as highlight genes when the log2FC were ≥1 or ≤−1, respectively.

### 2.7. Pathway Analysis

To facilitate the characterization of the overlap genes for ferroptosis, we used existing pathways based on the same RNA sequencing methods used in our paper [21]. We used Pathcards (https://pathcards.genecards.org/card/ferroptosis?, accessed on 30 August 2024) to assess the current status of ferroptosis pathways [22]. It was then modified and grouped according to specific genes within the “STRING Interaction Network for Ferroptosis” [23]. Finally, the pathway was customized in Adobe Illustrator (version 28.7.1) to color code genes according to the log2FC of our data set.

### 2.8. Human Brain Slices

The work on human brain tissue received ethical approval from the Friedrich Alexander University Ethics Committee (Vote Nr. 193-18B). Only patients with epileptic surgery (anterior temporal lobe resection) were included. All patients have given informed consent before surgery. No patient-specific information was gathered. Extralesional, apparently normal cortical tissue, which would otherwise have been discarded, was placed into ice-cold artificial cerebrospinal fluid (mM): NaCl 125, KCl 3, CaCl_2_ 2.5, MgSO_4_ 1.3, NaH_2_PO_4_ 1.25, NaHCO_3_ 26, Glucose 13 (EcoCyte Bioscience, Austin, TX, USA), which was oxygenated by gassing (95% O_2_/5% CO_2_). Transport took less than 15 min to the laboratory, where the tissue was immediately cut into 150 µm thick brain slices using a vibratome (Leica), as described before [24,25]. Human brain sections were transferred to perfusion chambers (Ibidi, Munich, Germany) and perfused constantly with oxygen- and carbonate-saturated aCSF (≈20% O_2_/5% CO_2_, flow rate ≈ 0.16 mL/min, pH 7.4) at room temperature, as described before [26,27]. The following substances were added to the aCSF, either alone or in combination: FeCl_3_ (200 µM), FAS (300 µM), deferoxamine (250 µM), and ferrostatin-1 (1 µM). Subsequently, the tissue slices were perfused for a period of 4 h, after which they were fixed in 4% PFA (Sigma-Aldrich) for 24 h and subsequently transferred to PBS.

### 2.9. Immunhistochemistry

Human brain slices were washed 3 times in phosphate-buffered saline (PBS, Sigma-Aldrich) and then blocked in 10% donkey serum (Merck)/0.3% Triton X-100 (Sigma-Aldrich) in PBS. Slices were incubated with primary antibodies for PDGFRβ (R&D Systems, AF385, 1:200) and GPX-4 (Sigma-Aldrich, MABS1274, 4 µg/mL) overnight. After washing 3 times in PBS, secondary antibodies (Dianova, Alexa Fluor 647 IgG donkey anti rabbit, LifeTechnolgies (Frederick, MD, USA), Alexa Fluor 488 IgG donkey anti mouse) were applied for 1.5 h and then again washed 3 times with PBS and incubated with DAPI (Merck (Darmstadt, Germany), D8417, 1:5000) for 2 min. After mounting, tissue slices were imaged on the ZEISS Axio Observer microscope and analyzed with the Zeiss ZEN Blue edition imaging program (version 3.6).

### 2.10. Statistical Analysis

Data sets were tested for Gaussian distribution followed by the appropriate statistical test. Data are presented as mean  ±  standard error of the mean (s.e.m.). *p* values were calculated using two-tailed unpaired Student’s *t* test, two-tailed Mann–Whitney test, one way ANOVA and Kruskal–Wallis test. Differences were considered statistically significant at *p* values below 0.05. All data, unless otherwise noted, were analyzed using GraphPad PrismTM V.9.3.1.

## 3. Results

### 3.1. Pericytes Are More Susceptible to Trivalent than to Divalent Iron

Human pericytes were incubated with either divalent iron (FAS) or trivalent iron (FeCl_3_). The WST analysis (Figure 1A) demonstrated that the exposure of cells to both trivalent (viability 32.8%, *p* < 0.0001) and divalent iron (viability 25.7%, *p* < 0.0001) for 24 h led to a reduction in cell viability compared to the control. The addition of deferoxamine, a chelator that binds exclusively to trivalent iron, was observed to result in the rescue of pericytes that had been treated with FeCl_3_ (viability 88.9%, *p* < 0.0001), but not with FAS (viability 31.9%, *p* = 0.9948). As demonstrated in Figure 1B, a duration of 1 h is sufficient to observe a significant effect in the WST analysis, a period of 2–4 h shows a reduction in cell viability up to 50%. Figure 1C demonstrates that concentrations of 200 µM FeCl_3_ and 300 µM FAS were needed to detect an effect on the cell viability of the pericytes.

In the FACS analysis (Figure 2), which demonstrates cell survival, comparisons were made not only between divalent and trivalent iron but also between the corresponding hemoglobin (Hb and Met-Hb). The FACS analysis revealed a markedly reduced survival of pericytes in the presence of FeCl_3_ (survival 78.5% vs. 94.3%; *p* < 0.0001) in contrast to the observations made in the presence of FAS (survival 87.1% vs. 94.3%; *p* = 0.2302) and various oxygenated hemoglobin (survival for Hb 95.0% vs. 94.3%; *p* = 0.9995; survival for Met-Hb 97.7% vs. 94.3%; *p* = 0.8795). The addition of deferoxamine was found to effectively rescue the detrimental impact of FeCl_3_ on pericyte cell death (survival 93.5% vs. 78.5%; *p* < 0.0001). The data presented in Figure 1 and Figure 2 suggest that the iron released during the breakdown of hemoglobin, particularly in its oxidized form (iron-III), appears to exert the greatest influence, limiting not only the function of pericytes (WST analysis) but also significantly contributing to pericyte cell death (FACS). In order to gain insight into the mechanism underlying the toxic effect of trivalent iron, we next conducted RNA whole-genome next generation sequencing on pericytes incubated with trivalent iron.

### 3.2. Trivalent Iron Leads to Increased Expression of Cell Stress Gene Clusters in Pericytes

There were clear differences in the expression patterns of the pericytes treated with iron (III) compared to the untreated control. After setting cut-offs (log2FC −0.5/+0.5, corrected *p*-value ≤ 0.01), the comparison of FeCl_3_ with the untreated control showed that 2013 genes were positively expressed and 2259 genes were negatively expressed out of a total of 60,666 genes examined. Thus, 7% of the pericyte genome expression was altered by FeCl3 exposure under these cut-offs. A cluster analysis was applied to the RNA sequencing data. The heat maps shown in Figure 3 and Figure 4 are a compilation of some of the most strikingly regulated gene clusters under FeCl_3_ exposure according to the external database we used for comparison (IPA). The most activated gene clusters in Figure 3 were subsumed under more global categories such as growth failure, cell sensitivity, and morbidity and mortality, implying that there is a general tendency toward a stressful state of pericytes under iron exposure. Most activated highlight genes were TMEM119, HMOX1, MGP, and MSC (Figure 3). The most strongly inhibited gene clusters were more specific and related to intracellular compartments and processes (Figure 4). These included the organization of the cytoskeleton, particularly microtubule dynamics, and the formation of protrusions with a large number of down-regulated genes that reach the cut-off (Figure 4). The pericyte cytoskeleton plays a critical role for the pericytes, as protrusions are essential for the maintenance and function of the blood–brain barrier [28]. This cluster analysis appears to be consistent with the results found in the WST cell viability essay that pericyte function is restricted under trivalent iron exposure.

### 3.3. Pathway Analysis Indicates That Pericytes Undergo Ferroptosis When Exposed to Trivalent Iron

The signaling pathway, which, according to our data, mainly occurs in pericytes following exposure to FeCl_3_, is illustrated in Figure 5 using pathway analysis (Pathcards.com, free online version) on our RNA sequencing set. Our findings revealed that GPX-4 and HMOX1 are critical players in this process, connecting the systems involved, including iron metabolism, oxidative stress, lipid metabolism, and the glutathione system. The signaling pathway identified by our pathway analysis is strikingly similar to a cell death signaling pathway known as ferroptosis, in which GPX-4 plays a key role, as well as lipid peroxidation [16,17], while other forms of cell death, such as necrosis, apoptosis or pyroptosis, which are characterized, in particular, by an emphasis on different caspase enzymes [29], showed no significant change. Figure 6 A shows the heat map of the significant genes in our data set associated with ferroptosis. The table below illustrates the up- or downregulated ferroptosis genes with the highest statistical significance, exceeding a certain log-fold2 change. Of these, GPX-4 exhibited one of the greatest significances (*p*-value < 1.08 × 10^−30^) with a log2fold change of 1.09 (Figure 4B). With regard to ferroptosis, mandatory for the glutathione system in the pathway analysis (Figure 5), the SLC7A11 transporter is significantly (*p*-value < 1.30 × 10^−4^) reduced in its expression under trivalent iron with a log2fold change of −0.39.

### 3.4. Pericytes Respond to Trivalent Iron Exposure with GPX4 Enrichment at Capillary Branching Sites

The importance of GPX-4 and the related assumption that ferroptosis plays a decisive role in the damage of pericytes by trivalent iron were investigated in the following experiment in ex vivo human tissue slices. During epilepsy surgery, marginally healthy human brain tissue was removed and tissue slices were perfused with 200 µM FeCl_3_ for four hours under near-physiological conditions. In the subsequent immunohistochemical analysis, pericytes were identified by their typical morphology, as described previously [28]. It was found that GPX-4 was significantly enriched exclusively at the branching sites of the capillaries (Figure 6C). This enrichment was significantly increased in tissue treated with FeCl_3_ and rescued in tissue treated with FeCl_3_ and deferoxamine (Figure 6D). Ferrostatin-1, the first small molecule found with the ability to inhibit ferroptosis, was also able to reduce GPX4 enrichment in pericytes at capillary branching sites (Figure 6C,D). This indicates that ferroptosis is one of the major pathways occurring in pericytes at branching sites after trivalent iron exposure.

### 3.5. Iron Exposure Leads to Reduced Overall Vascular Continuity and Tends to Have an Increased Vasoconstriction at Capillary Branching Sites

The length of fully connected capillaries in a broadly selected region of interest was found to be reduced under both 300 µM FAS and 200 µM FeCl_3_, indicating a disruption in vascular continuity (Figure 7J). The use of the trivalent iron chelator deferoxamine or the ferroptosis inhibitor ferrostatin-1 did not result in any detectable impact on the reduced vascular continuity (Figure 7J). During this analysis, it was observed that capillaries that were exposed to trivalent more than to divalent iron exhibited increased regions of vasoconstriction, which manifested initially at the branching sites and subsequently affected a portion of the adjacent capillary (Figure 7B,C). An attempt to quantify the amount of this vasoconstriction did not lead to significant results, so we can only speak of a remarkable trend here (Figure 7K). However, it appears that these local vasoconstrictions are less frequent when deferoxamine or ferrostatin-1 is administered concomitantly with trivalent iron exposure (Figure 7A–I,K).

## 4. Discussion

In this study, we showed that free iron induces ferroptosis as a specific type of cell death in human brain pericytes, contributing to blood–brain barrier dysfunction, disruption of vascular continuity and vasoconstriction. Scavenging free iron represents a therapeutic target to attenuate secondary injury after intracerebral hemorrhage, but several aspects deserve attention to assess intracerebral hemorrhage-associated iron release and its implications for future research.

Among the blood degradation products, free iron, but not hemoglobin, evoked detrimental effects on pericyte survival, which were more pronounced after exposure to trivalent than to divalent iron in this study. These findings might seem counterintuitive, as divalent iron is known to generate radical compounds such as reactive oxygen species (ROSs) which are harmful for cell metabolism [15,30]. However, recent studies found that binding to transferrin is required for the uptake of iron into the cell and only trivalent iron is bound by transferrin [31,32]. Therefore, divalent iron must oxidate prior to cell uptake and divalent intracellular iron originates from trivalent intracellular iron. Hemoglobin per se did not exert substantial toxic effects on cell function or survival, which is in line with previous publications additionally showing that hemoglobin is degraded to free iron accumulating in pericytes [15,33]. In a similar constellation as we find in subarachnoid hemorrhage, it has been shown that the vasodilator nitric oxide can be bound by released hemoglobin [34]. The reduction of nitric oxide then interferes with the endothelin metabolism of the pericytes, leading to calcium uptake and vasoconstriction [35]. High levels of calcium in pericytes can even induce cell death in a rigor state [26,36]. It has therefore been suggested that released hemoglobin plays a significant role in the formation of vasospasm and pericyte cell death via nitric oxide scavenging and the endothelin-1 system in subarachnoid hemorrhage, which is why a large series of clinical trials (CONSCIOUS-1, CONSCIOUS-2, CONSCIOUS-3, REACT) investigated the impact of the endothelin-1 receptor antagonist clozasentan on vasospasm and clinical outcomes in these patients [37,38,39,40]. We assume that the biochemical processes involved in the release of blood are similar in subarachnoid hemorrhage and ICH. Our experiments suggest that there are additional factors, beyond the scavenging effects of hemoglobin, which may be responsible for vasospasm and cell death. These additional factors could be the result of iron-induced toxicity, which could in turn explain the development of delayed secondary brain damage in intracerebral hemorrhage, particularly as the degradation of hemoglobin to free iron is supposed to take some time [41,42]. In accordance with our hypothesis, elevated concentrations of free iron have also been identified in traumatic brain injury involving blood release, in which ferroptosis has also been shown to play a significant role [41,43,44,45]. The finding that trivalent iron, but not hemoglobin, leads to ferroptosis-induced pericyte cell death is intriguing and suggests a possible target for future therapeutic design. 

To elucidate the underlying intracellular mechanisms, we used whole-genome RNA sequencing pathway analysis of trivalent iron-exposed pericytes and identified a specific form of programmed cell death, ferroptosis, characterized by intracellular iron overload leading to ROS generation [16,46]. Specific reductase systems such as glutathione (GSH) and glutathione peroxidase 4 (GPX-4) scavenge ROS and protect cells from DNA damage and lipid peroxidation, preventing cell membrane breakdown and cell death in ferroptosis [47]. In our experiments, TMEM119 was one of the highest activated genes when pericytes were exposed to iron (Figure 3). TMEM119 is known to activate ATF4, a regulator of metabolic and redox processes, which is important for the glutathione system, including GPX-4 [48]. HMOX1 was also upregulated when pericytes were exposed to iron (Figure 3, Figure 5 and Figure 6) and is known to be enriched in microglia and the perivascular spaces promoting secondary brain injury after intracranial hemorrhage [49,50]. The SLC7A11 transporter, a subunit of the system xc (-), which is responsible for the uptake of cysteine for the production of glutathione, was reduced in its expression when pericytes were exposed to trivalent iron [51]. Similar findings were observed in which cells underwent ferroptosis when treated with sorafenib, an inhibitor of the system xc (-) and pharmaceutical agent employed in the treatment of kidney and liver cancers [30]. Overall, there was a simultaneous activation of genes that induce the redox system due to damage processes and a strong indication that the glutathione system, and thus ferroptosis, plays an important role in cell death when pericytes are exposed to trivalent iron.

Since GPX-4 is highly involved in ferroptosis and in our RNA sequencing data, we evaluated the spatial distribution of GPX-4 among pericytes in the capillary bed using ex vivo human brain tissue slices. GPX-4 was elevated in pericytes, predominantly at capillary branching sites, when exposed to trivalent iron (Figure 6). These particular pericytes are known to regulate cerebral blood flow by vascular contraction and dying pericytes undergo a state of rigor, leading to the vasoconstriction of capillaries [26,36]. Accordingly, we observed clearly constricted capillaries at their branching sites when exposed to trivalent iron, and although these differences were not statistically significant in our study, vasoconstriction and consequently reduced blood supply represent a plausible secondary injury parameter in intracerebral hemorrhage. Furthermore, we found disrupted vascular continuity when human brain tissue slices exposed to trivalent iron, trivalent iron indicating disruption of the blood–brain barrier, aggravated edema formation and secondary injury after intracerebral hemorrhage [52].

### 4.1. Can Brain Damage by Iron and Ferroptosis Be Prevented?

The iron chelator deferoxamine effectively restored pericyte function and survival in our cell culture experiments (Figure 1 and Figure 2) and yielded positive results in nearly all in vitro experimental studies involving iron or blood degradation products [53,54,55], but was ineffective in our human brain tissue slice experiments. Similarly, deferoxamine did not significantly improve clinical outcomes of intracerebral hemorrhage patients in a large randomized clinical trial [7]. Besides trial design issues, deferoxamine barely crosses the blood–brain barrier after intravenous administration, which is why alternative pharmaceutical agents are needed. The ferroptosis inhibitor ferrostatin-1 showed beneficial effects on pericyte GPX-4 expression in our experiments and yielded positive results in brain hemorrhage mouse models [56,57], but had no effect on disrupted vascular continuity in our human brain tissue slice experiments. A possible explanation for the ineffectiveness of deferoxamine and ferrostatin-1 in rescuing human brain slices is the system’s inherent complexity compared to cell culture experiments, characterized by numerous influencing factors (e.g., cell–cell interactions, release of neurotransmitters, etc.) that remain to be elucidated. This work demonstrates the potential of new therapeutic targets in cell culture, but also highlights the challenges associated with application in complex systems. 

Besides dosage issues, further research is required to assess effective ferroptosis inhibitors and iron chelators, respectively, in clinical practice. Following positive ENRICH trial results, minimally invasive hematoma evacuation will become part of the standard of ICH care [58]. Complete removal of the hematoma ensures optimal clinical outcomes, but is hardly ever achieved [59]. Therefore, future studies should evaluate iron chelators of ferroptosis inhibitors administered in addition to surgical treatment strategies to effectively prevent secondary injury by blood degradation products of the residual hematoma and achieve optimal patient outcomes. Surgical interventions in turn reduce hematoma volume and blood degradation products and prevent the scavenging capacities of iron chelators or ferroptosis inhibitors to be exceeded, which is why surgical and iron binding strategies should exhibit synergistic effects. 

### 4.2. Limitations

Cell culture experiments were based on one line of human pericytes, which might limit generalizability. However, results were confirmed on ex vivo human brain slices, corroborating our findings. This study focused on the effects of blood degradation products on human pericytes as major components of the blood–brain barrier and capillary function, because pericytes are particularly susceptible to its detrimental effects [60], but future research should also address implications for further components of the human brain.

## 5. Conclusions

Here, we demonstrated for the first time that ferroptosis, a specific type of cell death, occurred in human brain pericytes when exposed to trivalent iron and may contribute to blood–brain barrier dysfunction, disruption of vascular continuity and increased vasoconstriction. Free iron and ferroptosis represent therapeutic targets to attenuate secondary brain injury in intracerebral hemorrhage. Future studies should evaluate iron chelators and ferroptosis inhibitors in addition to minimally invasive surgical treatment of intracerebral hemorrhage.

## Figures and Tables

**Figure 1 biomolecules-15-00164-f001:**
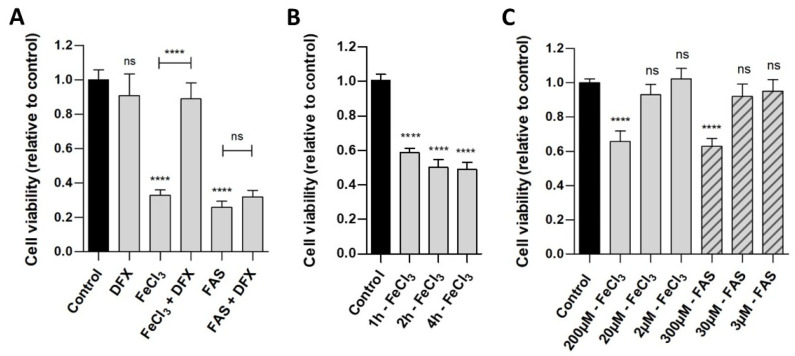
The effect of divalent and trivalent iron on pericyte cell viability. A standardized number of human pericytes were incubated with either divalent iron (FAS) or trivalent iron (FeCl_3_) for 1, 2, 4 or 24 h and cell viability was measured using the WST-1 assay. Cell function was analyzed in relation to untreated pericytes (control) and rescue was performed with trivalent iron chelator deferoxamine (DFX). (**A**): The exposure of cells to both trivalent (200 µM FeCl_3_) and divalent (300 µM FAS) iron for 24 h resulted in a marked and comparable reduction in cell viability. The addition of deferoxamine led to a rescue of pericytes treated with FeCl_3_, but not with FAS. (**B**): Following a period of one hour of incubation with 200 µM FeCl_3_, a significant reduction in cell viability was observed. Over a period of two to four hours, a further reduction in cell viability was seen, reaching a maximum of 50%. (**C**): The concentration series demonstrated that concentrations of 200 µM FeCl_3_ and 300 µM FAS were needed for detecting an effect on the cell viability of the pericytes after 4 h of incubation. With 2–20 µM FeCl_3_ and 3–30 µM FAS, no significant effect was detected after 4 h of incubation. All data are displayed as mean ± SEM and Dunn’s corrected Kruskal–Wallis test was used for multiple comparisons; ns = *p* > 0.05, **** *p* ≤ 0.0001; n = 6.

**Figure 2 biomolecules-15-00164-f002:**
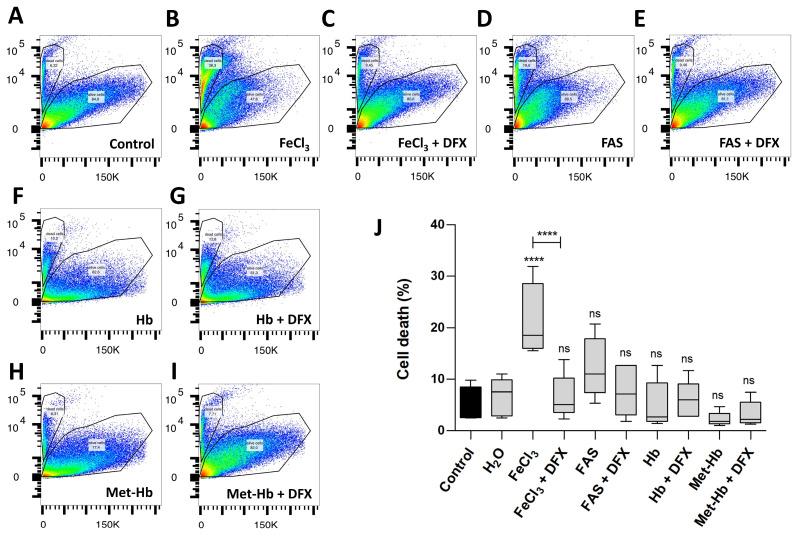
The effect of divalent and trivalent iron on pericyte cell survival (FACS). A standardized number of human pericytes were incubated with either divalent iron (FAS, 300 µM) or trivalent iron (FeCl_3_, 200 µM) or their corresponding hemoglobin (Hb, 9.3 µM and Met-Hb 9.3 µM) for 4 h and cell survival was measured using the FACS. Rescue was performed with trivalent iron chelator deferoxamine (DFX, 250 µM). (**A**–**I**): SSC-a versus PE-Texas Red diagram of the FACS analysis: dead and living pericytes shown as polygonal gates with divalent (FAS) and trivalent iron (FeCl_3_) in comparison to divalent (Hb) and trivalent (Met-Hb) hemoglobin with or without deferoxamine. (**J**): FACS analysis shows a highly significant increase in pericyte cell death under FeCl_3_, but not under FAS or various oxygenated hemoglobin. Deferoxamine was able to abolish the effect on pericyte cell death under FeCl_3_ exposure. All data are displayed as mean ± SEM and Dunn’s corrected Kruskal–Wallis test was used for multiple comparisons; ns = *p* > 0.05, **** *p* ≤ 0.0001; n = 6.

**Figure 3 biomolecules-15-00164-f003:**
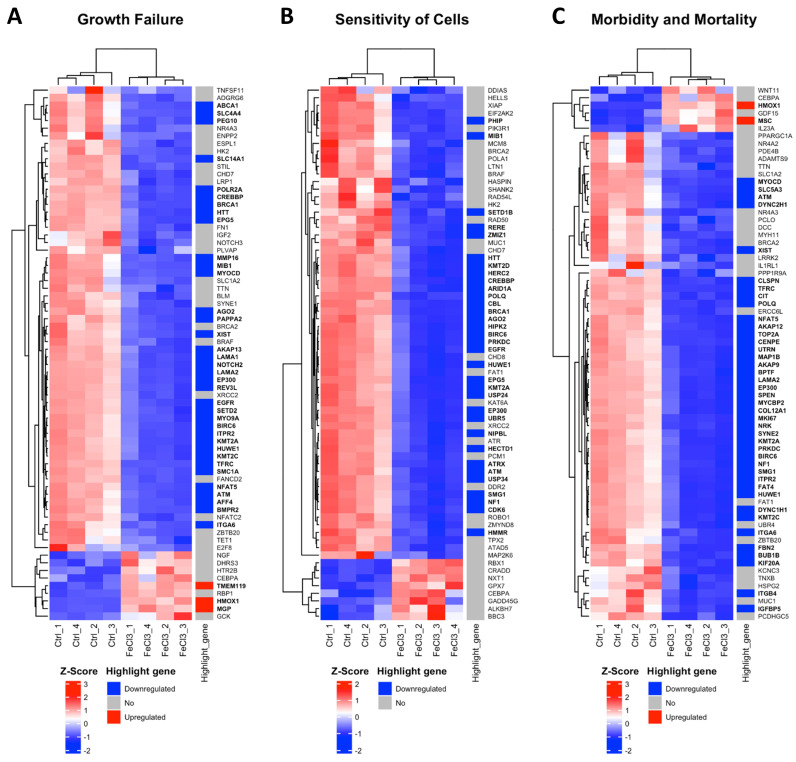
Most activated gene clusters in pericytes under trivalent iron exposure. RNA sequencing was performed and our data were compared with existing gene clusters from the Ingenuity Pathway Analysis database. The gene clusters are presented in the form of heat maps, which contain the majority of the genes that also exhibited a change in our data when human brain pericytes were treated with FeCl_3_ 200 µM for 24 h. The figure illustrates the three most prominent heat maps, which encompass the clusters of genes that exhibited the greatest activation in response to iron exposure. The up- or downregulation of a highlighted gene was marked with red for “upregulated” and blue for “downregulated”. The highlighted genes were labeled with a log2 fold change ≥1 or ≤−1, with a cut-off for *p* ≤ 10^−20^. The corresponding gene names are printed in bold.

**Figure 4 biomolecules-15-00164-f004:**
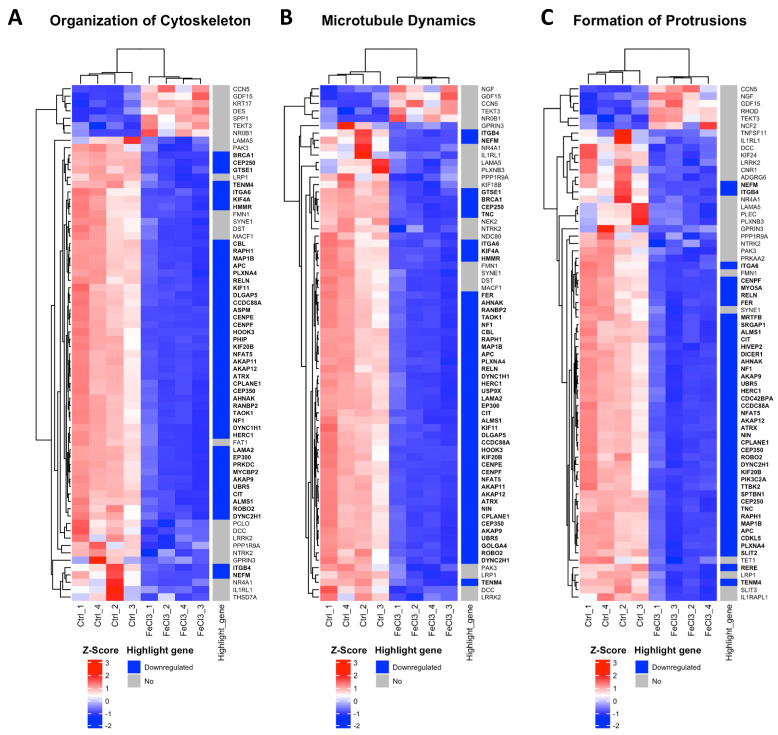
Most inhibited gene clusters in pericytes under trivalent iron exposure. RNA sequencing was performed and our data were compared with existing gene clusters from the Ingenuity Pathway Analysis database. The gene clusters are presented in the form of heat maps, which contain the majority of the genes that also exhibited a change in our data when human brain pericytes were treated with FeCl_3_ 200 µM for 24 h. The figure depicts the top three heatmaps, which illustrate the clusters of genes most significantly downregulated in response to iron exposure. The up- or downregulation of a highlighted gene was marked with red for “upregulated” and blue for “downregulated”. The highlighted genes are labeled with a log2 fold change ≥1 or ≤−1, with a cut-off for *p* ≤ 10^−20^. The corresponding gene names are printed in bold.

**Figure 5 biomolecules-15-00164-f005:**
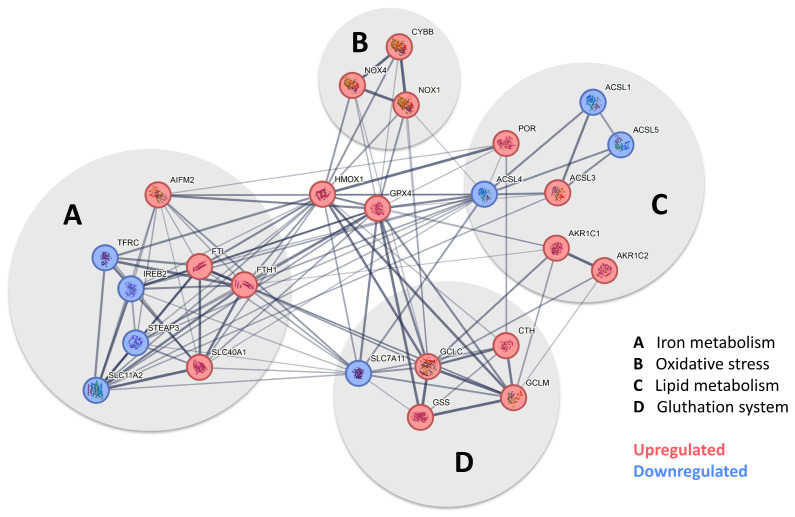
Pathway analysis indicates that pericytes undergo ferroptosis when exposed to iron. After the screening of the expression profile of our data set with available pathway data sets, a substantial overlap was identified with the ferroptosis pathway when human brain pericytes were exposed to trivalent iron. Figure 5 shows the pathway analysis tailored to our data, derived from the ferroptosis pathway currently available at https://pathcards.genecards.org/card/ferroptosis?, accessed on 30 August 2024. The major supergroups identified were iron metabolism, oxidative stress, lipid metabolism, and the glutathione system, marked with gray circles. Highlight genes reaching the previously defined cut-off were either more (red) or less (blue) expressed when pericytes were exposed to 200 µM FeCl_3_ for 24 h. The interconnection between the respective genes suggests the possibility of a potential interaction. The presence of a broad line indicates a relevant relationship previously described in the database. GPX-4 and HMOX1 were identified as key players in all depicted systems.

**Figure 6 biomolecules-15-00164-f006:**
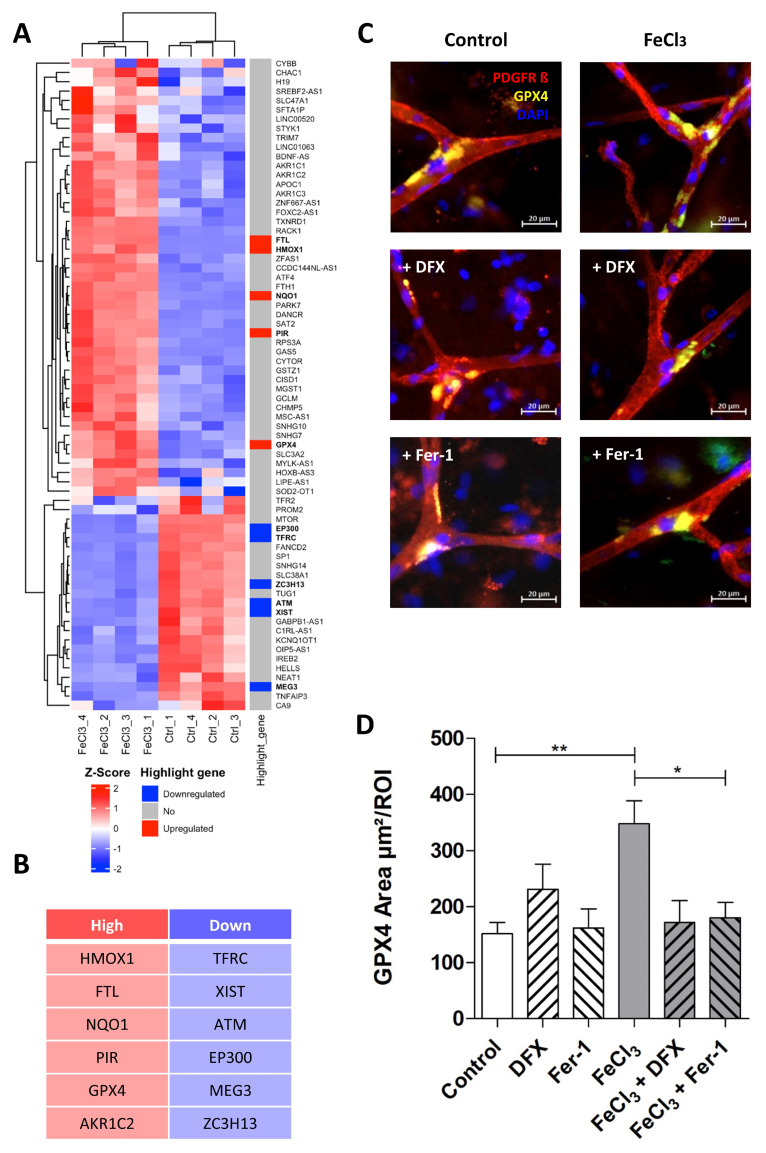
Trivalent iron leads to increased GPX-4 expression at capillary branching sites. The ferroptosis gene cluster, in accordance with our data, is presented in the form of a heat map with highlight genes pointed out (**A**). The top five signature genes that are upregulated (red) or downregulated (blue) in this specific cluster are listed below (**B**). We exposed human brain tissue ex vivo to 200 µM FeCl_3_ for 4 h and stained with the blood vessel and pericyte marker, PDGFRß, and GPX-4, one of the key players in ferroptosis. Rescue was performed with trivalent iron chelator deferoxamine (DFX). Pericytes were identified by their distinctive morphology, exhibiting a round and compressed nucleus stained with DAPI, in contrast to endothelial cells, which display a classical elongated nucleus (**C**). We found that the GPX-4 area at the branching sites of capillaries was significantly increased when exposed to trivalent iron, while concomitant treatment of ferrostatin-1, a ferroptosis inhibitor, effectively prevented this increase (**C**,**D**). All data are expressed as mean ± SEM. Multiple comparisons were conducted using Dunn’s corrected Kruskal–Wallis test; * *p* ≤ 0.05, ** *p* ≤ 0.01; n = 3.

**Figure 7 biomolecules-15-00164-f007:**
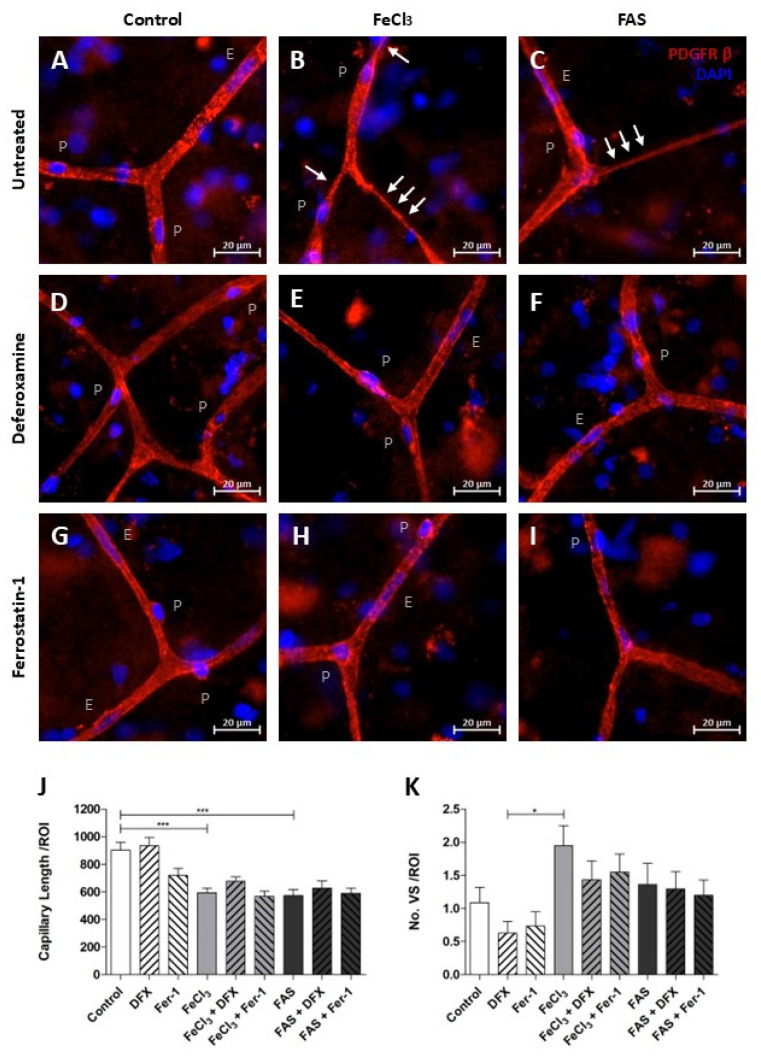
Iron exposure impairs the structural integrity of capillaries. Human brain tissue was exposed to 300 µM FAS or 200 µM FeCl_3_ for a period of four hours and stained with PDGFRβ and DAPI. Pericytes were then identified, as previously described. *p* = pericyte, E = endothelium cell, and white arrows illustrate vasoconstriction (**A**–**I**). Under divalent and trivalent iron exposure, the average capillary length per region of interest (ROI) is significantly reduced as a consequence of vessel disruption, but simultaneous treatment with neither deferoxamine nor with ferrostatin-1 led to an improvement in capillary continuity (**J**). The frequency of vasospastic vascular components per ROI was quantified, revealing a non-significant trend towards an increased prevalence of vasospastic areas in capillaries following trivalent iron exposure (**B**,**K**). A non-significant trend was also observed in the reduction in vasoconstriction when deferoxamine or ferrostatin-1 was administered concomitantly (**E**,**H**,**K**). All data are expressed as mean ± SEM. Multiple comparisons were conducted using Dunn’s corrected Kruskal–Wallis test; * *p* ≤ 0.05, *** *p* ≤ 0.001; n = 3.

## Data Availability

The raw data supporting the conclusions of this article will be made available by the authors on request. Raw sequencing reads and expression data for the RNAseq experiment are publicly available from NCBI GEO under the identifier GSE230184.

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
