# Peer review of "Intracerebral Hemorrhage-Associated Iron Release Leads to Pericyte-Dependent Cerebral Capillary Function Disruption"

_biomolecules, 2025, doi:10.3390/biom15020164_

Round 1
Reviewer 1 Report
Comments and Suggestions for Authors
Very well written manuscript. I have two minor recommendations:
1. The use of AE2S as a symbol in the text for ferrous ammonium sulfate is unusual and jarring. Consider using FAS instead or refer to it in the figures as Fe2+ so that it stylistically matches the Fe3+ which is used elsewhere.
2. SLC7A11 is frequently implicated in ferroptosis because of its role in cystine uptake. In this manuscript it appears that SLC7A11 is downregulated, however, there is no discussion about this transporter in the manuscript despite its canonical importance.
Reviewer 2 Report
Comments and Suggestions for Authors
1. The finding that Fe(III), but not hemoglobin induced ferroptosis is intriguing and suggests a possible target for future therapeutic design.
2. Cells were treated with 2, 20, or 200 uM FeCl3. What was the basis for choosing these three concentrations?
3. Why was 200 uM at 24 hours chosen for RNA seq when Fig 1B shows significant loss of cell viability at 1 hour (as well as 2 and 4 hrs)?
4. What was the cell viability at 200 uM for 24 hrs? Might this make it difficult to determine which RNA are participating in cell death and which are simply the result of advanced cell death processes?
5. Line 175 refers to divalent iron, but subsequent references are to bivalent iron.
6. All figures need more descriptive legends that better explains the graphs. For example, what was the iron concentration and time of exposure in Fig 1A? What was the concentration in Fig 1B? time frame in Fig 1C?
7. Very difficult to read gene names on heat maps.
8. Consider making figures 1 D-L a separate figure with its own descriptive legend.
9. It is not clear how the 3-4% estimate (line 223) is calculated. 2013+2259/60666=7%
1. Is it possible to speculate why deferoxamine and ferrostatin-1 were effective in vitro, but not in human brain slices? While it is true that deferoxamine does not readily cross the BBB, that does not explain the findings in brain slices.
Round 2
Reviewer 2 Report
Comments and Suggestions for Authors
The authors have addressed the questions raised in this review.
Author Response
According to the reviewer report form, all questions were adressed.